# A Hardware-Simplified Soft-Start Scheme for Current-Fed Full-Bridge DC-DC Converter

**Shupeng Li** [1], **Zhenbin Li** [1], **Weiqi Meng** [2] **and Jinwei He** [2,*]

[1] State Grid Tianjin Electric Power Research Institute, Tianjin 300392, China; shupeng.li@tj.sgcc.com.cn (S.L.); zhenbin.li@tj.sgcc.com.cn (Z.L.)

[2] School of Electrical and Information Engineering, Tianjin University, Tianjin 300072, China; weiqi_meng@tju.edu.cn

[*] Correspondence: jinwei.he@tju.edu.cn; Tel.: +86-182-02539408

**Abstract:** At present, the current-fed full-bridge DC-DC converter still faces two inherent defects, namely, the soft-start issue during the start-up process and the voltage spike issue in normal working conditions. In existing research, a large number of attempts have been reported to overcome these two inherent issues. However, the existing solutions are all based on both redundant soft-start circuits and snubber circuits, and two different circuits are required. As a result, the cost of the current-fed full-bridge DC-DC converter is high, especially because the soft-start circuits are only used during the start-up process. So, to reduce device redundancy, a hardware-simplified soft-start scheme is proposed in this paper. In the proposed scheme, the snubber circuit is reused for the soft-start process based on topology reconfiguration, avoiding additional soft-start circuits to reduce the cost of this DC-DC converter. In this proposed method, the current-fed full-bridge DC-DC converter is operated as the voltage-fed full-bridge DC-DC converter by always turning on the snubber switch during the start-up process. Then, the output capacitor can be charged by slowly increasing the output voltage of the H bridge. Then, when the output voltage is high enough, the current-fed full-bridge DC-DC converter is again operated in the current-fed mode to boost the voltage. Subsequently, the converter characters and control strategy were examined and analyzed. The experimental results verify the feasibility of the proposed hardware-simplified soft-start scheme for the current-fed full-bridge DC-DC converter.

**Keywords:** current-fed; soft-start; voltage spike; snubber circuit





## 1. Introduction

The current-fed full-bridge DC-DC converter has some unique advantages such as low current ripple, inherent short-circuit protection, high step-up ratio, and direct current controllability. Thus, it is widely used as an interface converter for renewable energy systems, etc. [1–3]. However, this type of converter has two intrinsic problems, namely, the soft-start issue during the start-up process and the voltage spike issue in normal working conditions [4–8]. A large number of studies have focused on overcoming these problems, but two different circuits should be required to overcome them. As a result, the cost of the current-fed full-bridge DC-DC converter is relatively high, especially when the soft-start-up circuit is only used during the start-up process [9]. Then, this soft-start-up circuit would be wasted, which is not reasonable [10].

It is well known that the current-fed full-bridge DC-DC converter is a kind of boost converter, so it inherits some characteristics from the original boost converter [11]. Importantly, during the start-up process, due to the natural characteristic of boost-type converters, the filter inductor current experiences an in-rush increase when the output voltage is lower than the input voltage [12]. As a result, the current-fed full-bridge DC-DC converter may be damaged because of large current stress and voltage spike. Hence, to reduce the current

stress and voltage spike during the start-up process, the output voltage should be established before the normal working condition. For the buck converter, a fundamental method is to charge the output capacitor gradually using a duty ratio below 0.5. However, the duty ratio in the current-fed full bridge is limited to a value above 0.5 to provide continuous paths for the filter inductor current. Furthermore, the characteristic of the current-fed full-bridge DC-DC converter is very different from that of the buck converter but the same as that of the boost converter. Thus, the soft-start-up method for the buck converter cannot be suitable for the current-fed full-bridge DC-DC converter. Generally, additional start-up circuits are required in current-fed full-bridge converters, such as flyback circuits [12–15]. Based on the flyback circuits, the output voltage of the current-fed full-bridge DC-DC converter can be pre-charged, and when the output voltage is established, the current-fed full-bridge DC-DC converter can be operated in normal working conditions. By contrast, the start-up process of the voltage-fed converter is easier, and no start-up circuit is required.

The current-fed full-bridge DC-DC converter faces another significant problem. Because this converter contains two inductances in the circulating path, namely, the dc-link inductance and the leakage inductance of the middle transformer, the connection of these two inductances is a challenge [16,17]. When the currents of these two inductances are different during the connecting process, the voltage spike will occur because of the current mismatch between the dc-link inductor and the leakage inductor during the turn-off actions of switches [16,17]. To deal with this issue, some snubber circuits have been proposed to suppress voltage spikes [12–15], which can reduce the voltage spikes during the connection of the dc-link inductor and the leakage inductor. Furthermore, some snubber-less schemes are presented in [16–19] based on all-active-switch topology, which cannot be used in the current-fed full-bridge DC-DC converter. Moreover, in unidirectional applications such as photovoltaic generation and fuel cell systems, the all-active-switch topology is redundant and inevitably increases the complexity and cost of power converters. Hence, to reduce the cost, the snubber circuits are preferred in these applications, and the current-fed full-bridge DC-DC converter with a diode bridge on the secondary side is adopted.

In summary, for the current-fed full-bridge DC-DC converter, the soft-start-up method and the snubber method are required to realize the soft-start-up process and the soft commutation without large current stress and large spike voltages. All existing solutions to the start-up problem are based on the addition of soft-start circuits, and snubber circuits are required for the unidirectional current-fed DC-DC converter during working conditions. However, this soft-start circuit is only employed during the start-up process, which will boost the cost of the current-fed full-bridge DC-DC converter. So, to eliminate the redundant soft-start circuit, a hardware-simplified soft-start scheme is proposed in this paper based on the reuse of the snubber circuit. In the proposed soft-start scheme, the topology first works as a voltage-fed full-bridge converter during the soft-start process based on the snubber circuit. After the output voltage is established, it is reconfigured as an active clamping current-fed full-bridge converter. As a result of the clamping of the snubber circuit, the voltage spike can be suppressed effectively.

The rest of this paper is organized as follows. The configuration and proposed soft-start scheme are elaborated in Section 2, and converter characteristics and control strategy are illustrated in Section 3. Then, the experimental results and conclusion are given in Sections 4 and 5, respectively.

## 2. Configuration and Soft-Start-Up Schemes

In this section, the current-fed full-bridge DC-DC converter is demonstrated and discussed, including the snubber circuit. This circuit is always required for this unidirectional converter. Then, the conventional auxiliary circuits for the soft-start-up process of the current-fed full-bridge DC-DC converter is discussed. In addition, the proposed hardware-simplified soft-start scheme is proposed, where this converter is first operated as the voltage-fed DC-DC converter and then as the current-fed DC-DC converter.

### 2.1. Current-Fed Full-Bridge DC-DC Converter

Figure 1 shows the schematic of the current-fed full-bridge converter featuring a snubber circuit to limit the voltage spikes during the working condition. The snubber circuit is formed by a capacitor $C_{cap}$ and a switch $S_{cap}$. Notably, the capacitor is employed to generate the initial current of the leakage inductance before connection to the dc-link inductance [20–22]. $V_{in}$ and $V_o$ are the input voltage and the output voltage, respectively. $V_{cap}$ is the voltage on $C_{cap}$. The high-frequency transformer (HFTR) has a turn ratio of $n$ and leakage inductor $L_{leak}$. $I_{in}$ and $i_{leak}$ are the currents through the filter inductor $L_f$ and leakage inductor, respectively. $d$ and $d_{cap}$ are the duty ratios of switches $S_{1\sim4}$ and $S_{cap}$, respectively. $D_1\sim D_4$ and $C_o$ are rectifier diodes and output capacitor.

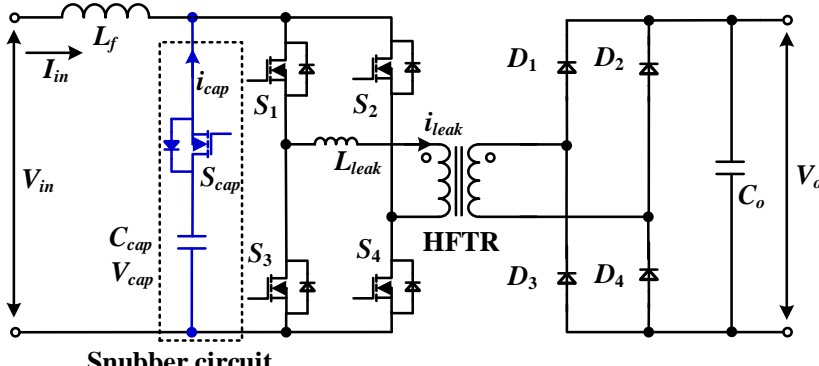

**Figure 1.** Schematic of the current-fed full-bridge converter with a snubber circuit.

In order to realize the power transmission of the current-fed full-bridge DC-DC converter, duty-ratio modulation is the most popular form of modulation, and a switching network is employed to connect the output voltage. So, the current-fed full-bridge DC-DC converter is very similar to the original boost converter in terms of the voltage characteristic. However, there are two inductances in this converter, namely, the dc-link inductance and the leakage inductance. So, the connection between these two inductances is not avoidable and, by combining the switch $S_{cap}$ and the capacitor $C_{cap}$, the snubber circuit can be used to realize the restrictions of the voltage spikes.

### 2.2. Conventional Auxiliary Circuits for Soft-Start-Up Process

To realize the soft-start-up process, some representatives of auxiliary circuits are introduced in Figure 2. Figure 2a presents a flyback snubber [12]. The flyback snubber clamps the voltage on switches to the voltage across the current-fed side of the transformer and recycles the absorbed energy in the clamping capacitor to the output side. The flyback snubber provides a continuous path to the inductor current; thus, the voltage spike can be avoided. In Figure 2b, external auxiliary buck circuits are utilized to achieve soft switching for the current-fed full bridge [13]. In Figure 2c, two passive capacitor-diode snubbers are added to the output-side switches [14]. The energy stored in the flyback snubber is transferred to the capacitor-diode snubbers; then, the voltage spike on the switches is reduced. In Figure 2d, a flyback soft-start circuit and a snubber circuit are utilized independently [15]. The snubber circuit clamps the voltage spike during the switching transition, and the flyback soft-start circuit recycles the energy absorbed in the clamping capacitor. Nevertheless, because these auxiliary circuits are only used during the short start-up process, the auxiliary circuits of all the above topologies result in redundant devices and complex control [23]. In addition, the cost of the current-fed full-bridge DC-DC converter is increased [24,25].

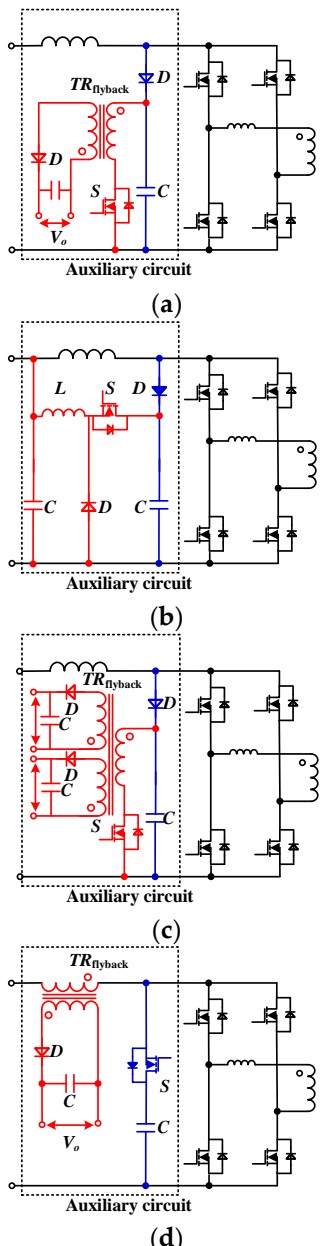

**Figure 2.** Schematic of various auxiliary circuits (Red part: soft-start circuit, Blue part: snubber circuit): (**a**) [12]; (**b**) [13]; (**c**) [14]; (**d**) [15].

### 2.3. Proposed Hardware-Simplified Soft-Start Scheme

The hardware-simplified soft-start scheme is proposed in this section. As shown in Figure 1, when the snubber switch is always turned on, the snubber capacitor voltage can be the same as the input voltage. Then, the slow increase in the output voltage of the H bridge can be employed to charge the output voltage. Thus, in this method, the current-fed full-bridge DC-DC converter is operated in two configurations, namely, the voltage-fed DC-DC configuration [26–28] and the current-fed DC-DC configuration [29–31]. Based on the capacitor of the snubber circuit, the current-fed full-bridge DC-DC converter can be operated in the voltage-fed mode. Then, the output voltage can be slowly pre-charged with the gradual increase in the duty ratio. When the output voltage is established, the current-fed full-bridge DC-DC converter is operated in the current-fed mode, and the snubber circuit is used to realize the soft connection of the dc-link inductance and the leakage inductance.

During the soft-start process, the capacitor in the snubber circuit is used to support the dc voltage on the input side of the current-fed full-bridge DC-DC converter, as shown in Figure 1. Then, the current-fed full-bridge DC-DC converter can be operated at the voltage-fed mode. Furthermore, the duty ratio $d$ can be increased slowly to avoid the in-rush current during the start-up process. The duty ratio $d$ of $S_1 \sim S_4$ increases linearly from 0.00 to the target value $d_{ref}$.

Figure 3 shows the theoretical waveforms of the proposed soft-start scheme in the range of 0.00~0.50 and 0.50~$d_{ref}$. The driving signals of $S_1$ and $S_4$, and of $S_2$ and $S_3$, are the same. $T_s$ is the switching cycle. As shown in Figure 3, the switch in the snubber circuit is always opened because the duty ratio $d_{cap}$ is equivalent to 1, and $d$, the duty ratio, is employed to provide voltage to the output side from the capacitor $C_{cap}$. Then, the current-fed full-bridge DC-DC converter acts as a buck converter. So, the output voltage of the H bridge can be slowly increased from 0 V, and the in-rush current can be omitted. Moreover, as shown in Figure 3, when the duty ratio $d$ is bigger than 0.5, there are some overlaps of the switches in the same half bridge, and the current-fed full-bridge DC-DC converter is transferred to the current-fed mode. The duty ratio $d$ is employed to regulate the output voltage. Furthermore, $d_{cap}$ is smaller than 1, and is employed to pre-charge the leakage inductance. Thus, when the dc-link inductance current is close to the leakage inductance current, the soft connection of the dc-link inductance and the leakage inductance can be realized.

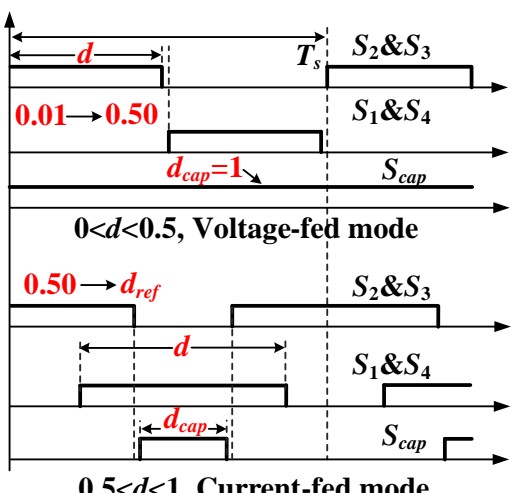

**Figure 3.** Theoretical waveforms.

Figure 4 shows the equivalent circuits in the range of 0.00~0.50. $S_{cap}$ stays open and $V_{cap}$ equals $V_{in}$. Although a soft-start method is also proposed in [32], the snubber switch is switched at twice the frequency of the switching frequency. Thus, the proposed method is simpler. In the proposed method, the input source $V_{in}$, inductor $L_f$, and snubber circuit can be seen as a voltage source. Then, the current-fed full-bridge DC-DC converter can be treated as a voltage-fed full-bridge DC-DC converter. Via this method, the output capacitor can be charged near the $n*V_{in}$ level. Then, the current-fed full-bridge DC-DC converter will be operated in the current-fed mode. Figure 5 shows the equivalent circuits in the range of 0.50~$d_{ref}$. In Figure 5a, $S_1 \sim S_4$ all conduct with an overlapping interval. In Figure 5b, $S_2$ and $S_3$ turn off. The inductor current charges the capacitor $C_{cap}$ through the body diode of $S_{cap}$, and the snubber circuit provides a continuous path to the inductor current. In Figure 5c, $S_{cap}$ turns on, and $C_{cap}$ recycles the stored energy to the output side.

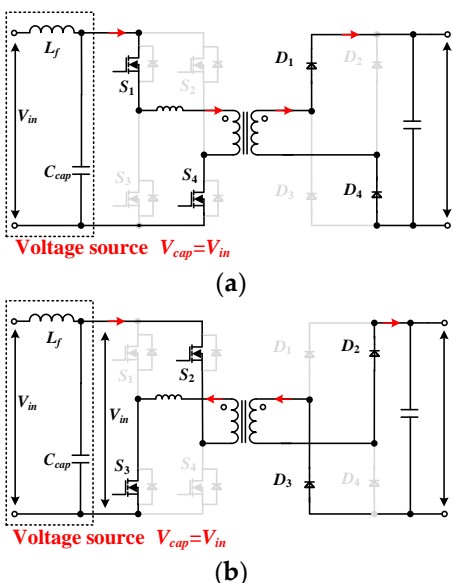

**Figure 4.** Range of 0.00~0.50 in voltage-fed mode: (**a**) $S_1$ and $S_4$ conduct; (**b**) $S_2$ and $S_3$ conduct.

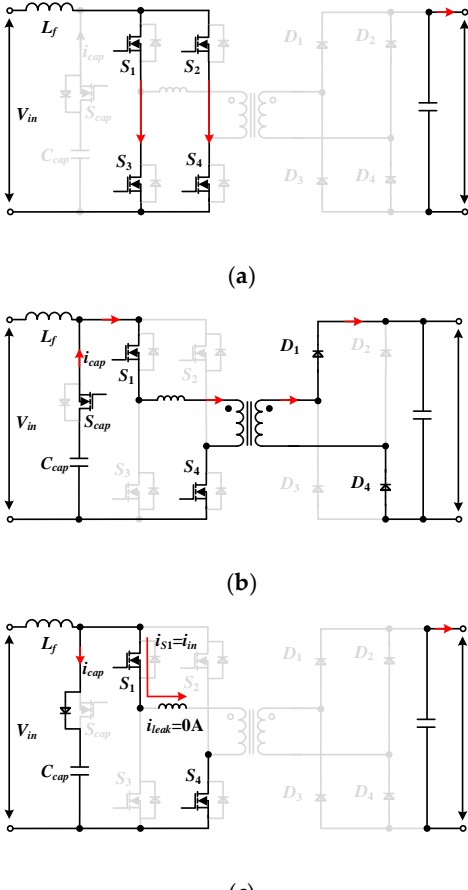

**Figure 5.** Range of 0.50~$d_{ref}$ in current-fed mode with voltage spike suppression. (**a**) $S_1$ ~ $S_4$ conduct; (**b**) $S_{cap}$, $S_1$, $S_4$ conduct; (**c**) $S_1$ and $S_4$ conduct.

## 3. Converter Analysis

In this section, the converter characteristics and the generation mechanism of the in-rush current during the start-up process are analyzed. Furthermore, the generation mechanism of the spike voltage during the working condition is analyzed. In addition,

the control analysis of the close-loop control is discussed, and the control method for the voltage-fed mode and the current-fed mode are presented.

### 3.1. The Analysis of Converter Characters

Figure 6 shows the generation mechanism of the in-rush start current. During the start-up process, there are two conditions of the current-fed full-bridge DC-DC converter: (1) the secondary side of the dc-link inductance is connected to the output voltage; and (2) the secondary side of the dc-link inductance is a short circuit. Furthermore, the output voltage is zero at the beginning of the start-up process. So, in either case, the inductor in-rush current will continue to increase because of the input voltage. Then, the large in-rush current will usually generate inductor saturation and damage the components. Thus, the soft-start process is necessary in a current-fed full-bridge converter, and the in-rush current should be reduced by the soft-start-up operation. Generally, the soft-start-up circuit is required to realize the soft start of the current-fed full-bridge DC-DC converter, and the operating modulation cannot meet this requirement.

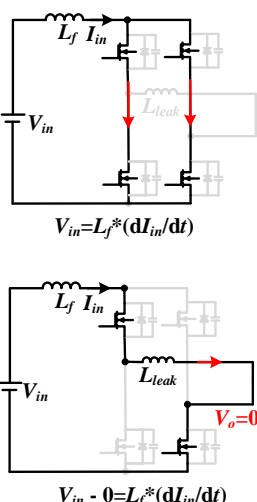

**Figure 6.** Generation mechanism of in-rush start-up current.

Figure 7 shows the generation mechanism of the voltage spikes. As mentioned before, there are two inductances in the circulating loop in the current-fed full-bridge DC-DC converter. So, the connection of these two inductances is required. Then, if the currents in the filter inductor and leakage inductor do not match but are directly connected during the switching off action, the switches on the current-fed side will suffer from high voltage spikes. The voltage spikes will usually damage the switch components. Thus, the snubber circuit is required for the current-fed full-bridge DC-DC converter.

The value of voltage spikes can be calculated using Equation (1) [33]. As shown in (1), the voltage spikes can be obtained via the parasitic capacitors of the switch $C_{oss}$, the time constant $\omega$, the leakage inductance $L_{leak}$, and the original currents ($I_{in}(0)$ and $i_{leak}(0)$). For example, when the parasitic capacitors of switches $S_1 \sim S_4$ are 80 pF (C3M0060065K), the relationship between the voltage spike, the different input currents $I_{in}$, and the leakage inductor $L_{leak}$ can be shown as Figure 8. As shown in this figure, the instantaneous voltage value can reach a level that is thousands of times the rated voltage. Then, the switches will be damaged by these voltage spikes. Thus, a snubber circuit is necessary for a current-fed full-bridge converter to avoid voltage spikes.

$$V_{spi}\left|\begin{array}{l} I_{in}(0) > 0 \\ i_{leak}(0) > 0 \end{array}\right. = \frac{(I_{in}(0) - i_{leak}(0))\frac{\sqrt{L_{leak}^3 L_f} + \sqrt{L_{leak} L_f^3}}{\sqrt{2C_{oss}(L_{leak} + L_f)}}}{L_{leak} + L_f}\sin(\omega t) \qquad (1)$$

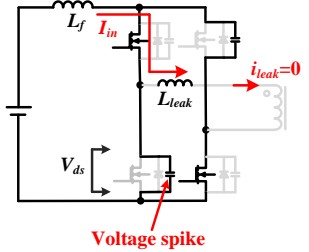

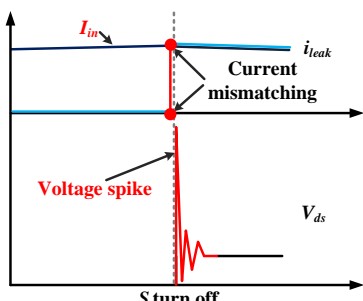

**Figure 7.** Generation mechanism of voltage spikes.

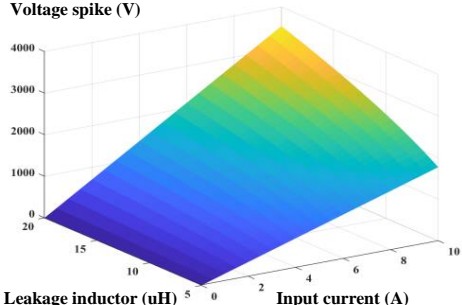

**Figure 8.** Voltage spike variation.

The converter characteristics are the same as those of an isolated buck converter in soft-start mode and an isolated boost converter in normal boost mode. According to the voltage-second balance of filter inductor $L_f$, the converter conversion ratio can be derived as Equation (2) [15,16]. Then, the theoretical target value of duty ratio $d$ at the rated working point is given in Equation (3), and the turn ratio of the output voltage can be drawn as Figure 9.

$$\begin{cases} L_f \frac{dI_{in}}{dt} = V_{in} & 0 \le t \le dT_s \\ L_f \frac{dI_{in}}{dt} = \frac{V_o}{n} - V_{in} & dT_s < t \le T_s \end{cases} \quad V_o = \frac{nV_{in}}{2(1-d)} \tag{2}$$

$$d = 1 - \frac{2V_o}{nV_{in}} \tag{3}$$

Meanwhile, the voltage stresses across switches $S_1 \sim S_4$ and $S_{cap}$ are expressed as Equation (4). The clamping capacitor $C_{cap}$ can be selected based on the resonant tank formed by $C_{cap}$ and $L_{leak}$. The resonance happens during the on-stage of $S_{cap}$ in boost mode operation. Hence, the criterion is that the resonant period should be longer than half the switching cycle, and the range of $C_{cap}$ is given in Equation (5) [12,15].

$$\begin{cases} V_{s2} = \frac{V_o}{n} & V_{Scap} = 0 & \text{start} \\ V_{s2} = \frac{V_{in}}{2(1-d)} & V_{Scap} = \frac{V_{in}}{2(1-d)} & \text{boost} \end{cases} \tag{4}$$

$$C_{cap} > \frac{(T_s/4\pi)^2}{L_{leak}} \tag{5}$$

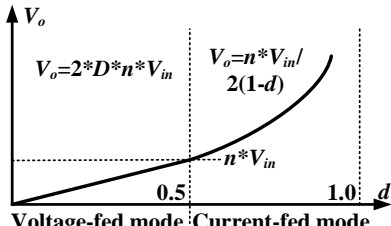

**Figure 9.** Turn ratio of $V_o$ versus duty ratio $d$ in the soft-start process.

### 3.2. Analysis of the Control Strategy

To realize the hardware-simplified soft-start scheme and the close-loop control, the control structure of the current-fed full-bridge DC-DC converter is illustrated in Figure 10, which has three steps to realize the proposed methods. Furthermore, the output voltage and the input current are employed to realize this proposed hardware-simplified soft-start scheme including the close-loop operation.

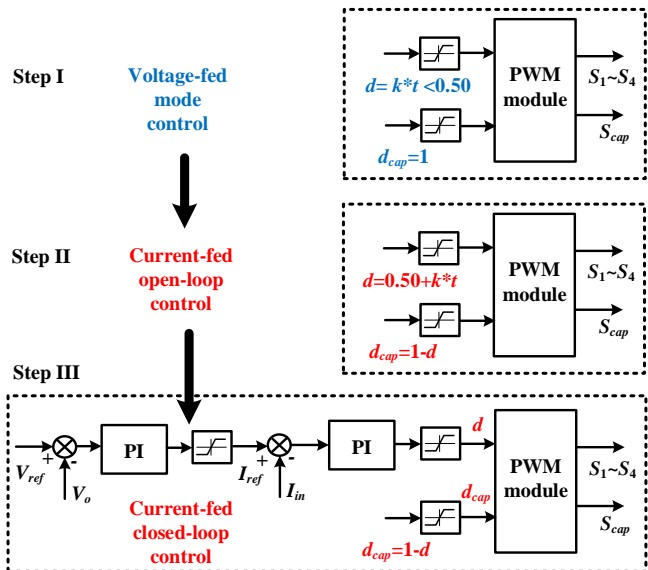

**Figure 10.** Control structure of the proposed soft-start scheme.

As shown in Figure 10, in step I, the current-fed full-bridge DC-DC converter is configured in the voltage-fed mode, and the duty ratio $d$ increases linearly from 0.00 to 0.50. The voltage-fed mode is utilized to establish an initial output voltage at the beginning of the start-up process. When the duty ratio $d$ reaches 0.50, the output voltage $V_o$ reaches close to $n*V_{in}$, and the converter is switched to the current-fed mode. Furthermore, the current-fed full-bridge DC-DC converter is operated in the current-fed mode as step II. Then, in step II, the duty ratio $d$ continues to rise linearly from 0.50 to the target value $d_{ref}$, which can be calculated as Equation (3). When the duty ratio $d$ reaches the target value $d_{ref}$, the output voltage is close to the desired value, and the soft-start process is able to enter step III. In this step, a dual-loop PI controller is utilized to regulate the output voltage, with the outer voltage loop and the inner current loop [11]. $V_{ref}$ and $I_{ref}$ are the reference values of output voltage and input current, respectively. In order to realize the close-loop control, the output voltage and the input current are measured. Then, based on the desired output voltage $V_{ref}$ and the output voltage $V_o$, the desired input current $I_{ref}$ can be obtained through the PI controller. Further, by combining the desired input current $I_{ref}$ and the input current $I_{in}$, the duty ratio can be obtained through the inner PI controller. Furthermore, duty ratio $d_{cap}$ is employed to control the switch of the snubber circuit.

In summary, there are three steps in the proposed hardware-simplified soft-start scheme for the current-fed full-bridge DC-DC converter, namely, the voltage-fed control

mode, the current-fed control mode, and the close-loop control mode. In the first two steps, the current-fed full-bridge DC-DC converter is operated in open-loop control mode. When the output voltage is close to the desired output voltage, the control scheme is switched from the open-loop control mode to the closed-loop control mode.

## 4. Verification

For this section, a scaled-down laboratory prototype was built to verify the performance of the proposed scheme. The HFTR is an EE55 Ferrite material core for the transformer. Switches $S_{cap}$ and $S_1 \sim S_4$ are CREE C3M0060065K, driven by CGD15SG00D2. Diodes $D_1 \sim D_4$ are ROHM SCS220AE. The digital controller is a dSPACE MicroLabBox DS1202. The voltage/current sensors are LEM LV25/LA25. Table 1 shows the main parameters for the platform of the current-fed full-bridge DC-DC converter, and Figure 11 shows the experimental waveforms to verify the soft-start process and voltage spike suppression.

**Table 1.** Parameters for the current-fed full-bridge DC-DC converter.

| Parameter | Value |
|---|---|
| Power range | 0~400 W |
| Switching frequency | 50 kHz |
| Leakage inductance | 8 μH |
| Filter inductor | 300 μH |
| Output capacitor | 200 μF |
| Rated input voltage | 100 V |
| Rated output voltage | 200 V |
| Transformer turn ratio | 1:1 |
| Snubber capacitor | 40 μF |

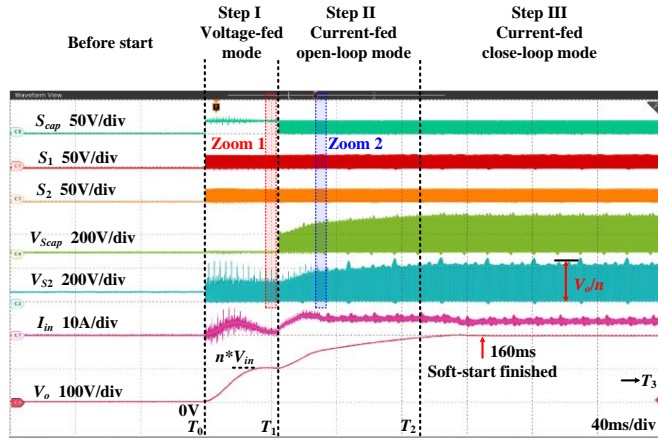

**Figure 11.** Experimental waveforms to verify the soft-start process and voltage spike suppression.

As shown in Figure 11, there are three steps in the proposed hardware-simplified soft-start scheme for the current-fed full-bridge DC-DC converter. During step I ($T_0 \sim T_1$), the converter operates in voltage-fed mode, and the duty ratio $d$ increases linearly with open-loop control. At the instant $T_1$, the duty ratio $d$ reaches 0.50, and the output voltage $V_o$ reaches close to $n*V_{in}$. Notably, the voltage spikes $V_{s2}$ of switch are caused by the hard switching, and are smaller than double the input voltage $U_{in}$. During step II ($T_1 \sim T_2$), the converter is switched to current-fed mode from voltage-fed mode. The output voltage $V_o$ continuously increases. During step III ($T_2 \sim T_3$), the closed-loop controller intervenes in controlling the duty ratio $d$, and finally, the output voltage is stable at the reference value. During the whole soft-start process, the drain-source voltage $V_{S2}$ on the current-fed

side switches (e.g., $S_2$) is always within a safe area $V_o/n$, consistent with the analysis in Section 2.3 The time taken for the soft start is about 160 ms, and is affected by the filter inductor value and the speed at which the duty ratio increases. Finally, the hardware-simplified soft-start scheme can be realized for the current-fed full-bridge DC-DC converter.

Moreover, Figure 12 shows the experimental waveforms of Zoom I. As shown in this figure, the magnified waveforms for the voltage-fed control mode can be obtained, where the phase-shift modulation method is employed. The snubber switch $S_{cap}$ is always turned on to damp the current increase in filter inductance $L_f$. Figure 13 demonstrates experimental waveforms of Zoom 2, and the typical waveforms for the current-fed modulation can be obtained. The oscillation is caused by the inherent hard switching of the current-fed converter, which is generated by the leakage inductance and the parasitic capacitor of the switches.

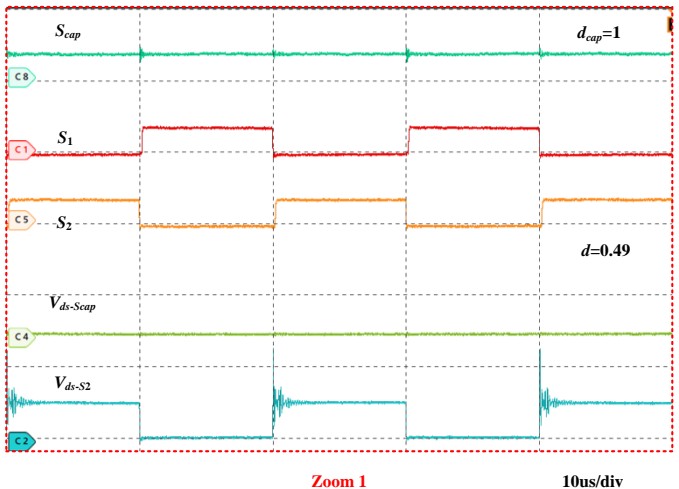

**Figure 12.** Experimental waveform with voltage-fed configuration.

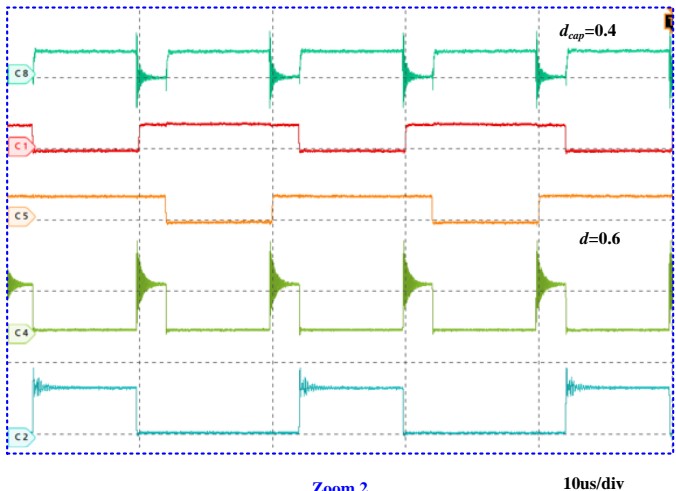

**Figure 13.** Experimental waveform with current-fed configuration.

Table 2 gives the comparison of existing soft-start methods [12–15,32] for current-fed full-bridge converters. As shown in this table, compared with the existing methods, the soft-start-up circuit in the proposed hardware-simplified soft-start scheme is not required for the current-fed full-bridge DC-DC converter. So, under the proposed method, the number of auxiliary circuits in the device can be reduced significantly, and only two components are needed to realize the soft-start-up operation and the voltage-spike snubber for the current-fed full-bridge DC-DC converter. In other existing schemes [12–15], at least five components are required, and the cost would be higher than that of the proposed schemes.

Furthermore, with fewer components, the control complexity of the proposed hardware-simplified soft-start-up method can be reduced obviously since fewer switches need to be operated. Although the soft-start-up method in [32] can eliminate the additional circuits, the operation of the switching signals is more complicated than in the proposed method. Thus, compared with these existing methods, the advantages of the proposed hardware-simplified soft-start scheme are obvious for the current-fed full-bridge DC-DC converter.

**Table 2.** Comparison of existing soft-start methods.

| Items | [12] | [13] | [14] | [15] | [32] | Proposed Method |
|---|---|---|---|---|---|---|
| Soft-start circuit | Need | Need | Need | Need | No need | No need |
| Device number of auxiliary circuit | 6 | 6 | 8 | 5 | 2 | 2 |
| Control complexity | High | High | High | High | Middle | Low |

In addition, the auxiliary circuit in [14] can also be employed to realize the near soft-switching performances of the switches and the efficiency should be higher. Furthermore, the auxiliary circuits in [12,13,15] also mainly focus on the soft-start-up operation of this converter, and more components usually generate higher power losses compared with the proposed method for the current-fed full-bridge DC-DC converter.

## 5. Conclusions

In this paper, based only on the snubber circuit, a hardware-simplified soft-start scheme is proposed for the current-fed full-bridge converter, where the soft-start-up circuit can be omitted. Based on the support capacitor in the snubber circuit, the basic principle of the proposed scheme is to start the converter as a reconfigured voltage-fed converter. After the establishment of the output voltage, the reconfigured voltage-fed converter is transitioned to the current-fed converter, and the snubber circuit continuously suppresses the voltage spike using the snubber circuit. The detailed procedure of the proposed soft-start scheme is discussed, and experimental results demonstrate superior performance. Compared with all existing soft-start methods for the current-fed full-bridge converter, the proposed soft-start scheme has the following advantages:

(1) In the existing operations for realizing the soft-start-up process and the suppression of voltage spikes, the soft-start circuit and the snubber circuit are both required for the current-fed full-bridge DC-DC converter. In contrast, only the snubber circuit is required in the proposed hardware-simplified soft-start scheme. Based on the reconfiguration of the circuit, the current-fed full-bridge DC-DC converter can be operated in voltage-fed mode and current-fed mode, and the voltage-fed mode can be employed to realize the soft-start-up process.

(2) Compared with the existing methods for the current-fed full-bridge DC-DC converter, fewer devices in the auxiliary circuits are required in the proposed hardware-simplified soft-start scheme. Thus, the modulation and the control for the proposed scheme can be reduced obviously, and the proposed scheme is simpler in terms of operation.

(3) This paper presents a new way to realize the soft-start-up operation of the current-fed full-bridge DC-DC converter with a snubber circuit, and the snubber circuit is employed to realize the soft connection of the dc-link inductance and the leakage inductance. Based on this principle, the proposed hardware-simplified soft-start scheme can also be extended to other current-fed DC-DC converters with a similar configuration.

**Author Contributions:** S.L. provided the funding and wrote the paper, Z.L. proposed the hardware-simplified soft-start scheme, W.M. provided the experimental results, and J.H. supervised this manuscript. All authors have read and agreed to the published version of the manuscript.

**Funding:** This research was funded by [State Grid Tianjin Electric Power Company] grant number [SGTJDK00DWJS2300316].

**Data Availability Statement:** Data is contained within the article.

**Conflicts of Interest:** The authors declare no conflict of interest.

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
