# Peer review of "A Hardware-Simplified Soft-Start Scheme for Current-Fed Full-Bridge DC-DC Converter"

_electronics, doi:10.3390/electronics13010180_

Round 1

Reviewer 1 Report

Comments and Suggestions for Authors

Introduction has already provided an overview of the conventional methods, and chapter 2.2 is again presenting the conventional methods. Please integrate the content of chapter 2.2 into the introduction.

From chapter 2, it is better to focus on explaining the proposed method.

The proposed method appears to lack novelty as it shares similarities with already well-established approaches.

e.g.) Van-Dai Bui, Honnyong Cha, Thien-Dung Tran, Duc-Tuan Do, “Soft-start Active-clamp Isolated Full-bridge Current-fed DC-DC Converter,” 2023 International Symposium on Electrical and Electronics Engineering (ISEE), 2023.

Please, introduce the methods that are similar to the proposed method, highlighting the key differences and advantages.

Author Response

Thank you for taking time to review our manuscript! This existing method is different from ours. In the existing method, the snubber switch is switched as double of the switching frequency, while the snubber switch is always turned on in the proposed method. More discussion and response can be obtianed in the attached file. Please check it. Thanks a lot!

Reviewer 2 Report

Comments and Suggestions for Authors

The article is undoubtedly relevant.

This article introduces a hardware-simplified soft-start scheme for the current-fed full-bridge converter, focusing on the snubber circuit. The proposed scheme eliminates the need for a separate soft start-up circuit. It operates by initially configuring the converter as a voltage-fed converter using the support capacitor in the snubber circuit. After establishing the output voltage, it transitions to a current-fed converter, with the snubber circuit continuously suppressing voltage spikes. The scheme requires only the snubber circuit, offering advantages over existing methods, such as reduced device count for auxiliary circuits, simpler modulation and control, and a novel approach to soft start-up. The proposed scheme's applicability is extended to other current-fed dc-dc converters with similar configurations. Experimental results confirm its performance compared to existing soft-start methods.

For what frequency range of operation is the proposed method suitable?

Author Response

Thank you for taking time to review our manuscript. The discussion and response can be found in the attached file. Please check it. Thanks a lot!

Reviewer 3 Report

Comments and Suggestions for Authors

The paper titled "A Hardware-Simplified Soft-Start Scheme for Current-Fed 2 Full-Bridge DC-DC Converter" deals with an important issue of this type of converter architecture, that of the start up to reach electrically stable operating condition, and as a consequence the avoidance/suppression of the overvoltage caused by an inductive current-fed circuit.

The paper is quite complete and well describes the various architectures and solutions proposed in the past.

I have just two main comments on it:

1) In Sec. 2.3 we read "The hardware-simplified soft-start scheme is proposed in this part, which is directly based on the snubber circuit." It is better to describe how the proposed is derived, taking into consideration the many examples of Figure 2.

2) The proposed snubber is non dissipative, i.e. it is active and the only dissipation term is in the conduction losses of the active switch. It would be advisable to spend a few words for the reader on the various snubber circuits shown in Figure 2, their impact on efficiency, and if they are also used to improve the power quality of the converter.

Comments on the Quality of English Language

Just minor problems.

Author Response

Thank you for taking time to review our paper. The detialed discussion and response can be found in the attached file. Please check it. Thanks a lot!

Reviewer 4 Report

Comments and Suggestions for Authors

Overall, this paper is well organized and the proposed method can effectively solve the soft-start issue and voltage spike issue. Two suggestions are given as follows:

1) Please show the picture of the prototype.

2) Please give more explanations regarding the mechanism of the proposed method.

Author Response

Thank you for taking time to review our paper. More response and discussion can be found in the attached file. Please check it. Thanks a lot!

Reviewer 5 Report

Comments and Suggestions for Authors

The soft start-up operation is always a problem to the current-fed dc-dc converters. To solve this issue, this paper proposed a simple soft start-up method by reusing the circuit of the snubber circuit for reducing the design cost, which is ingenious. Compared with the existing solutions, the advantages are obvious. So, I have only a small question: when the leakage inductance is very small, is the proposed method suitable for the start-up operation of the current-fed full-bridge dc-dc converter? 

Author Response

Thank you for taking the time to review our manuscript. More reponses can be found in the attached file. Please check it. Thanks a lot!

Reviewer 6 Report

Comments and Suggestions for Authors

This paper focuses on an innovative hardware-simplified soft-start scheme for current-fed full-bridge DC-DC converters, potentially advancing power conversion reliability in energy systems and electrical engineering applications.

1. The paper would benefit from a thorough review of language and grammar to enhance readability and professional presentation. 

2. I recommend expanding the literature review to include recent and relevant studies.

3. The unique contributions of your study are not sufficiently emphasized. Clarify how this work advances the field of DC-DC converters, enhancing the abstract and conclusion to distinctly outline the novelty and practical implications of your findings.

4. To improve clarity and accessibility of the experimental setup, present the parameters settings in a tabular format.

Comments on the Quality of English Language

The paper would benefit from a thorough review of language and grammar to enhance readability and professional presentation.

Author Response

Thank you for taking time to review our paper. More responses can be found in the attached file. Please check it. Thanks a lot!

Reviewer 7 Report

Comments and Suggestions for Authors

In this paper a hardware-simplified soft-start scheme is proposed for current-fed full-bridge dc-dc converter, which is interesting and well-written. There are some comments.

1. The name of Fig. 13 and Fig. 14 should be clearer, such as adding the detailed name of the processes.

2. As shown in Fig. 12, why there are some spikes in the input current Iin?

Comments on the Quality of English Language

English is overall good.

Author Response

Thank you for taking time to review our manuscript. More responses can be found in the attached file. Please check it. Thanks a lot!

Round 2

Reviewer 1 Report

Comments and Suggestions for Authors

Manuscript is modified.